# EXPLANATORY MASKS FOR NEURAL NETWORK INTERPRETABILITY

**Lawrence Phillips [1,*], Garrett Goh [2], Nathan Hodas [1],**

[1] Computing & Analytics Division, Pacific Northwest National Lab
[2] Advanced Computing, Mathematics & Data Division, Pacific Northwest National Lab
lawrence.phillips@pnnl.gov

## ABSTRACT

Neural network interpretability is a vital component for applications across a wide variety of domains. One way to explain neural networks is to indicate which input data is responsible for the decision via a data mask. In this work, we present a method to produce explanation masks for pre-trained networks. Our masks identify which parts of the input are most important for accurate prediction. Masks are created by a secondary network whose goal is to create as small an explanation as possible while still preserving predictive accuracy. We demonstrate the applicability of our method for image classification with CNNs, sentiment analysis with RNNs, and chemical property prediction with mixed CNN/RNN architectures.

## 1 INTRODUCTION

Network interpretability remains a required feature for machine learning systems in many domains. A great deal of recent work attempts to shed light on deep learning models, producing methods that create local explanations (Ribeiro et al., 2016), follow gradients (Selvaraju et al., 2016), perturb inputs (Fong & Vedaldi, 2017), and even generate textual explanations (Hendricks et al., 2016). Many of these techniques involve creating an input *mask*, assigning weights to individual inputs such that a weighted visualization of the input can be created.

In this work, we explore an alternate method for generating such a mask. Our goal is to generate an *explanatory* input mask with two properties: 1) the explanation should be as minimal as possible, 2) the explanation should highlight the portions of the input most necessary to ensure predictive accuracy. We demonstrate a variety of regularization techniques to minimize the mask and achieve predictive accuracy by creating masked inputs which can still be used by the original network.

## 2 EXPLANATION MASKS

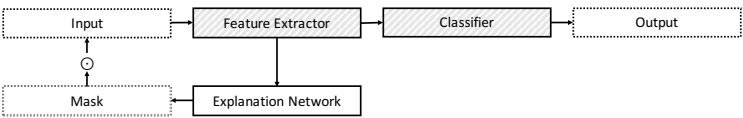

Figure 1: Explanation mask architecture. The pre-trained network (top) is divided into a feature extraction and classification component. The output of the feature extractor feeds into an explanation network, producing a mask that is multiplied element-wise with the input. The masked input is fed through the original network to produce a final output. Shading indicates frozen weights.

Given a pre-trained network, the goal of the explanation mask is to identify which inputs are most necessary for creating accurate decisions. To do this, we make use of a secondary, *explanation* network, which learns to generate explanation masks in an end-to-end fashion. By training the network end-to-end, we ensure the generated mask carries predictive accuracy. Further, by implementing regularization over the mask, we ensure that the mask is as minimal as possible while still being

accurate. This trade-off between accuracy and the size of the mask is one which can be explored on a per-case basis by changing the amount of regularization imposed.

The structure of the explanation network can be of any form fit to the problem task which leaves the method quite flexible. A schematic of the overall system is given in Figure 1. First, the pre-trained network is split into a feature extraction and classification component. The output of this feature extraction step is then fed into the explanation network. Given the features describing the input, the explanation network generates a mask of the same size as the input. The mask should have weights between 0 and 1 and is multiplied element-wise against the original input.

Once the input has been masked, it can then be fed through the entirety of the original network, both feature extraction and classification components, to produce a final output. To train the explanation network, we freeze the weights of the original network. This ensures that the explanation mask is an accurate reflection of what inputs are necessary for the pre-trained network. Training is accomplished end-to-end through the standard techniques such as backpropagation.

## 3 EXPERIMENTS

### 3.1 IMAGE CLASSIFICATION

We use the CIFAR10 image classification dataset (Krizhevsky & Hinton, 2009) to test our model's ability on visual data. As a base model, we use ResNet164 v2 (He et al., 2016a) which is a 164-layer residual network. The model achieves test accuracy of 94.04%.

As input to the explanation network, we use the 256-length vector ResNet generates before a final softmax decision. The vector is reshaped into a $16 \times 16$ matrix and passed through 4 conv. blocks. Each block has 4 2D conv. layers with ELU activations and 64 filters. Padding ensures the size of the image remains constant. A residual connection (He et al., 2016b) sums the first and final conv. layer activations. After the first block, the mask is upsampled to 32, the size of the image. After the final block, the explanation network ends with a single 2D convolution with filter size 1 and sigmoid activation.

The mask is multiplied by the original input and applied equally across all color channels. Sparsity is encouraged through $L_1$ and $L_2$ regularization. In Figure 2 we present representative examples.

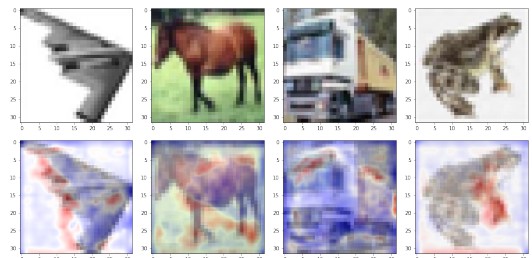

Figure 2: CIFAR10 images (top) along with their learned explanation masks (bottom).

### 3.2 SENTIMENT ANALYSIS

We also test our approach on text classification with the IMDB review sentiment task (Maas et al., 2011). We preprocess the input with a vocab size of 20,000 and maximum length of 500 words. Our base model uses 100-dimensional, pre-trained GloVe embeddings with 40% word-level dropout to prevent overfitting. This is followed by a bidirectional GRU layer of width 256. The output of this layer is averaged over timesteps and passed to a 128-dimension fully connected layer with ReLU activation and a final dense softmax. We optimize binary crossentropy with Adam. After 10 epochs we achieve a test accuracy of 84.7%.

Our explanation network uses the full output of the bidirectional GRU layer as input. We use a 2-layer bidirectional GRU of width 100. Each timestep's output is passed to a dense layer of size 256 with ReLU activation before a sigmoid layer outputs a single value, representing the word's mask

this is one of the worst movies i've ever seen it's supposed to be a remake or update of the one armed swordsman by chang the ham fisted direction and crappy fight choreography mean that

please give this one a miss kristy swanson and the rest of the cast rendered terrible performances the show is flat flat flat i don't know how michael madison could have allowed this one on his

i must be getting old because i was riveted to this movie from the first time i saw it i'm watching it again right now on hbo it's a very simple film about 2 people that fall in love after they found out

Figure 3: IMDB reviews with words highlighted based on explanation mask weights..

| **Positive** | best | great | very | though | well | music | real | still | think | most |
|---|---|---|---|---|---|---|---|---|---|---|
| **Negative** | boring | waste | worst | awful | terrible | horrible | poor | worse | bad | stupid |

Table 1: Top 10 positive and negative words in the IMDB explanation mask.

weight. We regularize by penalizing mask entropy. After 10 epochs we achieve a test accuracy of 81.8%. Figure 3 presents example sentences with words highlighted by mask weights. In Table 1 we also present the most attended words from positive and negative reviews.

### 3.3 CHEMICAL SOLUBILITY

As a final test, we demonstrate our method on a regression task with small data. We predict chemical solubility on the ESOL dataset (Wu et al., 2018), which contains only 1,128 data points.

Our base model is a mixed CNN-GRU network which takes in chemical compounds as SMILES strings (Weininger, 1988) and predicts solubility. The network is structured as in Figure 4a and when trained achieves a test RMSE of 0.846. The explanation network takes as input the embedded SMILES string and outputs a character-level explanation mask. The network is a 20-layer conv. residual network with SELU non-linearities and padding. The final layer is a 1D conv. of size 1 with batch normalization followed by a softplus activation. We regularize with $L_1$=1e-3 and $L_2$=1e-4 loss and after training reach a test RMSE of 0.831.

In Figure 4b, we highlight the explanation mask, with darker circles indicating atoms that are most important. For soluble molecules, the network properly attends to atoms which are in hydrophilic groups, which increase solubility. The opposite is true of insoluble molecules, as in the bottom right, where the network attends to the Cl molecule as part of a hydrophobic group.

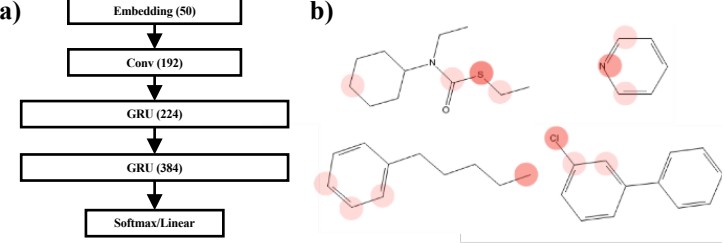

Figure 4: (a) Illustration of the SMILES2vec CNN-GRU architecture. (b) The explanation mask of SMILES2vec validates established knowledge by focusing on atoms of known hydrophobic and hydrophilic groups, colored circles of increasing darkness indicate increasing attention.

## 4 CONCLUSION

Through experiments over three diverse domains and network architectures we have shown how pre-trained networks might be analyzed by means of network-generated explanation masks. Although generating explanation masks requires a small amount of training for the explanation network, the method is flexible in its structure and because it is trained end-to-end is capable of identifying which parts of the input are most important for accurate classification.

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
