# OpenReview forum: "Explanatory Masks for Neural Network Interpretability"
_ICLR.cc/2018/Workshop — Reject_

### Official Review · AnonReviewer1 · 2018-03-06
**comparison with other network interpretability work**

**Rating:** 4
**Confidence:** 5

**Review:**

* Overview *
The paper proposes to predict the relevant parts of an input (e.g. image, text) that correlated with the output prediction. The authors train a NN to predict a mask which then they use to mask the input. The idea is simple however the authors fail to compare to similar works that deal with the same topic (see below)

* Details *
The authors should compare and contrast their work to
a) Visualizing and understanding convolutional networks, Zeiler & Fergus, ECCV 2014
b) Interpreting Deep Visual Representations via Network Dissection, Zhou et al, arXiv:1711.05611, 2017

I would like the authors to mention whether the performance of the classifier drops after masking the input. Also, the authors should mention why to train a network from scratch rather than interpreting the weights and activations of the layers instead, which does not involve training a new NN which results in changed weights.

---

### Official Review · AnonReviewer3 · 2018-03-10
**Advantages of proposed method are unclear**

**Rating:** 3
**Confidence:** 5

**Review:**

The description of the proposed method lacks detail.
The advantages of the proposed method over explanation techniques such as Sensitivity Analysis, Guided Backprop or LRP are unclear. Qualitative and quantitative comparison with other explanation methods is lacking.

---

### Official Review · AnonReviewer2 · 2018-03-13
**Simple but Interesting**

**Rating:** 7
**Confidence:** 4

**Review:**

In this work, the authors propose a simple explanatory mask generation network which given some feature representation portion of a deep network, outputs an attention map on the input which is sparse and which highlights the regions of the image that are minimally required to predict the correct output for the now-masked input. They apply this technique in three highly different domains (image classification, sentiment analysis, and chemical solubility) and produce reasonable results in each. There is limited novelty over standard attention mechanisms however the fact that this approach produces these maps in pixel space for frozen network is sufficiently interesting.

---

### Decision · Program_Chairs · 2018-03-20
**ICLR 2018 Workshop Acceptance Decision**

**Decision:**

Reject

**Comment:**

Based on the reviews, this paper has not been accepted for presentation at the ICLR workshop. However, the conversation and updates can continue to appear here on OpenReview.